

# Intermittent theta burst stimulation facilitates functional connectivity from the dorsal premotor cortex to primary motor cortex

Hai-Jiang Meng[1,*], Na Cao[2,*], Jian Zhang[2] and Yan-Ling Pi[3]

[1] School of Sports, Anqing Normal University, Anqing, China
[2] School of Psychology, Shanghai University of Sport, Shanghai, China
[3] Shanghai Punan Hosptial of Pudong New District, Shanghai, China
[*] These authors contributed equally to this work.

## ABSTRACT

**Background**. Motor information in the brain is transmitted from the dorsal premotor cortex (PMd) to the primary motor cortex (M1), where it is further processed and relayed to the spinal cord to eventually generate muscle movement. However, how information from the PMd affects M1 processing and the final output is unclear. Here, we applied intermittent theta burst stimulation (iTBS) to the PMd to alter cortical excitability not only at the application site but also at the PMd projection site of M1. We aimed to determine how PMd iTBS–altered information changed M1 processing and the corticospinal output.

**Methods**. In total, 16 young, healthy participants underwent PMd iTBS with 600 pulses (iTBS600) or sham-iTBS600. Corticospinal excitability, short-interval intracortical inhibition (SICI), and intracortical facilitation (ICF) were measured using transcranial magnetic stimulation before and up to 60 min after stimulation.

**Results**. Corticospinal excitability in M1 was significantly greater 15 min after PMd iTBS600 than that after sham-iTBS600 ($p = 0.012$). Compared with that after sham-iTBS600, at 0 ($p = 0.014$) and 15 ($p = 0.037$) min after iTBS600, SICI in M1 was significantly decreased, whereas 15 min after iTBS600, ICF in M1 was significantly increased ($p = 0.033$).

**Conclusion**. Our results suggested that projections from the PMd to M1 facilitated M1 corticospinal output and that this facilitation may be attributable in part to decreased intracortical inhibition and increased intracortical facilitation in M1. Such a facilitatory network may inform future understanding of the allocation of resources to achieve optimal motion output.

Corresponding author
Yan-Ling Pi, 18930502138@163.com

## INTRODUCTION

After a period of noninvasive brain stimulation or certain patterns of activity, the function as well as the very structure of the cortex of the brain changes. These functional and structural cortical changes occur through a process known as cortical plasticity (*Pascual-Leone et al., 2005*). Fundamental processes, such as motor learning, depend on neuronal plasticity occurring in a number of brain regions, especially among those areas that are spatially interconnected (*Luber & Lisanby, 2014*). However, recruitment of the functional connectivity that makes up the crucial interneuron networks among these interconnected regions differs (*Nettekoven et al., 2014*).

A growing body of research has shown that a broad set of key brain regions, known as the motor system network, are functionally involved in motor performance and motor control. This motor system network is not restricted to the motor cortex but dynamically extends into parietal, temporal, and prefrontal areas, depending on task complexity and an individual's experience (*Swinnen & Wenderoth, 2004*). The major functional area of the motor system, the dorsal premotor cortex (PMd), is crucial for controlling the preparation and execution of motor behaviors (*Roth et al., 1996*). The corticocortical connections from the PMd to the ipsilateral primary motor cortex (M1) are thought to transmit information relevant to generate the final motor output (*Koch et al., 2007*). However, how the PMd–M1 neural circuits achieve this output remains poorly understood, particularly after dynamic modifiability by external perturbation.

Methodological advances in noninvasive brain stimulation techniques—such as transcranial magnetic stimulation (TMS) and repetitive TMS (rTMS)—have informed our knowledge of cortical plasticity and its underlying mechanisms (*Ni et al., 2011*). Single-pulse TMS activates interneurons, and this activation discharges corticospinal neurons, generating motor-evoked potentials (MEPs) in the target muscle that can be used to assess the level of corticospinal excitability (*Hallett, 2007*). A different type of TMS protocol, called paired-pulse TMS, can be used to probe intracortical circuits that are highly interconnected and to determine the final motor cortical output. Short-interval intracortical inhibition (SICI), a well-investigated intracortical inhibitory phenomenon, can be elicited when a subthreshold conditioning stimulation suppresses a subsequent suprathreshold test stimulation (at interstimulus intervals of 1–5 ms), leading to inhibition of the subsequent MEP. Intracortical facilitation (ICF) is elicited with a similar protocol but using longer interstimulus intervals of 6–30 ms (*Kujirai et al., 1993*). Studies have reported that the balance and interaction between intracortical inhibition and facilitation are critical for the regulation of neuronal excitability and plasticity (*Dai et al., 2016*). Intermittent theta burst stimulation (iTBS), a type of rTMS, alters motor cortex excitability. This altered excitability has been shown by an increase in the amplitude of MEPs measured at the innervated target muscle for as long as 20 min following single-pulse TMS (*Cardenas-Morales et al., 2010*; *Huang et al., 2005*). In addition, rTMS is capable of evoking aftereffects at distant sites (such as M1) that are interconnected with the stimulated cortex (such as PMd). These remote aftereffects have been attributed to effective activation of output and input connections between the PMd and M1 (*Bestmann et al., 2003*; *Rizzo et al., 2004*). Thus, the results of
such studies demonstrate that PMd and M1 interconnections can be assessed by TMS measurement of cortical excitatory changes in M1.

The application of rTMS not only influences the characteristics of neurons within the stimulated region but also may affect activity levels of remote but interconnected areas, and this remote change is a long-lasting aftereffect (*Bestmann et al., 2003*). For example, a study using TMS combined with magnetic resonance imaging showed that TMS with anterior–posterior directed current applied to M1 strengthens the connectivity between the premotor and M1 regions (*Volz et al., 2015*). These stimulation-induced changes in M1 are attributable to both local factors and the interconnections with the premotor region (*Cardenas-Morales et al., 2014*). In addition, a previous study using TMS combined with electroencephalography showed that cerebellar TBS modulated cortical excitability of distant interconnected motor areas through common temporal, spatial and frequency domains (*Casula et al., 2016*). These findings suggest that a relationship exists between the responsiveness to rTMS and the motor network connectivity (*Koch et al., 2007*).

In the present study, we investigated whether the aftereffects induced by iTBS on the PMd affected the excitability of the ipsilateral M1 and its intracortical circuits. We hypothesized that the aftereffects induced by a specific iTBS protocol would increase the excitability of M1 and would decrease the degree of the intracortical inhibition. We further hypothesized the existence of a facilitatory network from the stimulated PMd to the ipsilateral M1.

## MATERIAL AND METHODS

### Participants

We studied 16 right-handed, healthy participants (8 women and 8 men; mean age, $22.19 \pm 1.72$ years) with no history of neurological or psychiatric diseases. Right-handedness was determined based on the Edinburgh Handedness Inventory (*Oldfield, 1971*). All participants provided written informed consent in accordance with the Declaration of Helsinki. The protocol was approved by the ethics committee of the Shanghai University of Sport (reference No. SUS2014024).

### Electromyographic recording

To record electromyogram signals from a selected target muscle, the first dorsal interosseous (FDI) muscle of the right hand, Ag-AgCl surface electrodes (nine mm in diameter) were placed at the tendon belly. The signal was bandpass filtered (20 Hz to 2.5 kHz) and then amplified 1,000 times (Intronix Technologies amplifier, model 2024F). The signal was then digitized at 5 kHz using a Micro1401 data acquisition unit (Cambridge Electronics Design, Cambridge, UK). The resulting data were analyzed with Signal software, version 6.02.

### Transcranial magnetic stimulation

We generated a monophasic current that traveled in a posterior to anterior direction in the left hemisphere M1 by using coils (9.5 cm in diameter) with a figure-eight shape that were connect to a stimulator (Magstim 200²; Whitland, Dyfed, UK). The coil handle was held so that is was approximately 90° to the central sulcus and at a 45° angle to the central sagittal line. The coil was moved until a suprathreshold stimulation of the left hemisphere

M1 generated the highest MEP amplitude in the target muscle, the right FDI. The position of the coil was then marked.

The measurements recorded included the resting motor threshold (RMT), MEP amplitude (a measure of the degree of corticospinal excitability), SICI, and ICF that were induced with a paired-pulse TMS paradigm (*Ni et al., 2011*). We defined RMT as the minimum stimulation necessary to induce MEPs at the FDI that had amplitudes measured from peak to peak of at least 50 μV in 5 or more of 10 trials while the FDI muscle was relaxed. To evaluate iTBS-induced MEP amplitude changes, the amplitude was compared with that before iTBS using a 1-mV TMS intensity. This 1-mV intensity was defined as the lowest TMS intensity needed to generate MEPs with amplitudes greater than 1 mV in 5 or more of 10 trials while the FDI muscle was relaxed. For the SICI and ICF paired-pulse TMS paradigms, they were consisted of a subthreshold condition pulse set as 70%RMT and a suprathreshold test pulse set at 1 mV intensity and were adjusted as needed at each time point to maintain the amplitude of test MEPs at approximately 1mV, but left the conditioning intensity unchanged (*Huang et al., 2005*; *Ni et al., 2014*). The interstimulus interval was 2 ms for SICI and 10 ms for ICF. The interstimulus interval of 2ms for SICI was selected because initial testing of intervals from 1 to 5 ms indicated that the MEP amplitude showed peak suppression at this time (*Kujirai et al., 1993*). Similarly, the interstimulus interval of 10 ms for ICF was selected because 10 ms is the most effective interval to induce ICF (*Ni et al., 2011*). Ten trials for each measurement (i.e., MEP amplitude, SICI and ICF) were conducted in a random order.

## Theta burst stimulation

The present study followed a previously described iTBS protocol with a slight modification (*Huang et al., 2005*). Briefly, one burst was defined as three pulses given at a frequency of 50 Hz. We gave 10 bursts every 200 ms for 2 s. Ten more bursts were given every 10 s for 191 s to generate a total of 600 pulses; thus, this protocol is termed iTBS600 herein. To prevent preactivation of the FDI muscle, we decreased the intensity of the iTBS used in the original protocol (which was 80% of the individual's active motor threshold) to 70% of the RMT (*Gentner et al., 2008*; *Huang et al., 2008*). Previous studies using this modification have reported aftereffects consistent with studies using the original stimulation (*Gentner et al., 2008*; *Cardenas-Morales et al., 2014*).

In the present study, TBS stimulation was delivered to the left hemisphere PMd. The PMd was defined as the area 2.5 cm anterior and one cm medial to the left hemisphere M1$_{Hand}$ area measured from the scalp (*Mochizuki, Huang & Rothwell, 2010*).

## Experimental design

To investigate how cortical excitability and plasticity reorganization affect the motor system network after iTBS is applied to the PMd, we conducted two experimental sessions for each participant: iTBS600 applied to the PMd, and sham-iTBS600 applied to the PMd. The sham-iTBS600 protocol consisted of sham stimulation blocks. To reduce potential cortical stimulation effects for these sham sessions, the coil was held at 45°, with the rim opposite the handle of the coil—rather than the center of the coil—touching the skull.

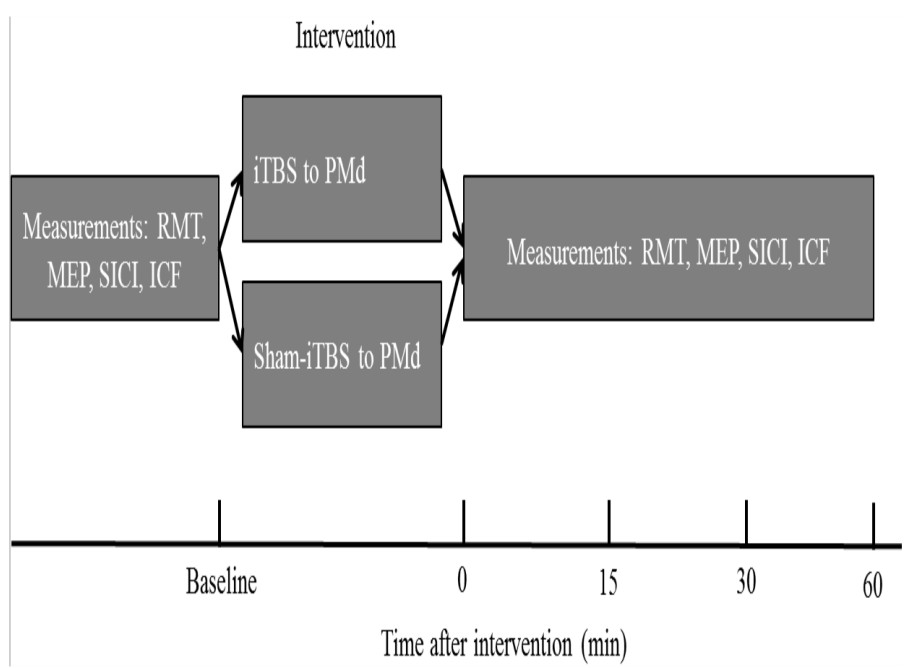

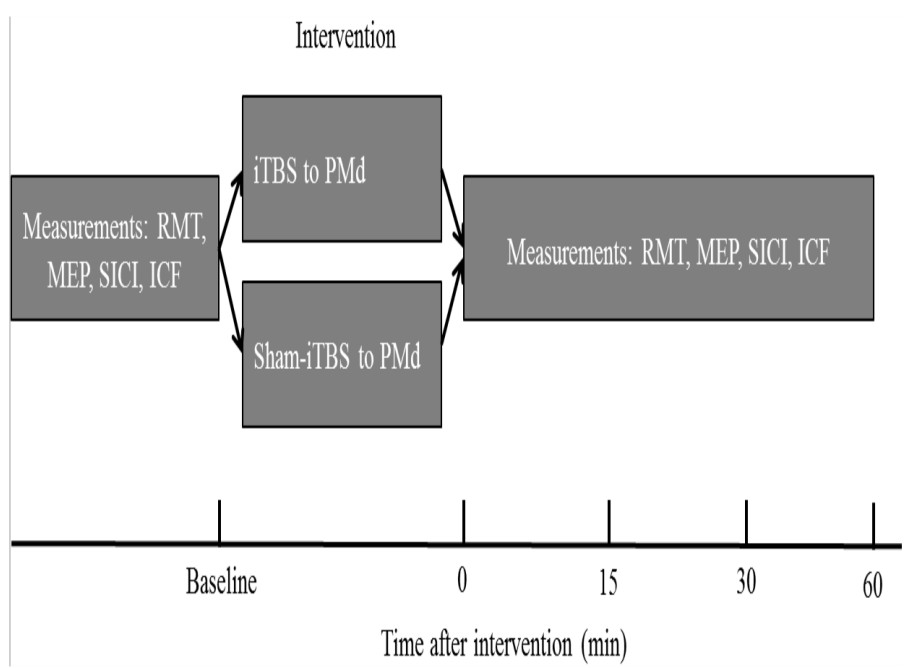

**Figure 1  Schematic of the experimental design.** The four measurements, resting motor threshold (RMT), motor-evoked potential (MEP) amplitude, intracortical facilitation (ICF), and short-interval intracortical inhibition (SICI), were obtained before (at baseline) as well as immediately and up to 60 min after application of the interventional protocols. iTBS indicates intermittent theta burst stimulation, and PMd represents dorsal premotor cortex.

Previous studies have reported that this is an effective sham stimulation method because the coil–cortex distance is relatively large, substantially reducing the electromagnetic field (*Cao et al., 2018*).

The order of the sessions in which either PMd iTBS600 or PMd sham-iTBS600 was applied was counterbalanced among the participants. Participants were asked to wait at least 2 weeks between the first and the second experimental sessions to avoid potential confounding of the results. The experimental design is shown in Fig. 1. Measurements were taken before (baseline) as well as 0, 15, 30, and 60 min (T0, T15, T30, T60) after each interventional protocol.

### Data and statistical analyses

To examine the effects of iTBS600 or sham-iTBS600 and time on the MEP amplitude and on SICI or ICF, we used two-way repeated-measures analyses of variance (ANOVAs). Protocols (iTBS vs sham-iTBS) and time (baseline, T0, T15, T30, T60) were within-subject factors included in the ANOVAs. Significant findings were further probed with post hoc Bonferroni tests, which corrected for multiple comparisons.

All statistical analyses were conducted using SPSS software, version 17.0, and two-sided values of $p < 0.05$ indicated statistically significant differences. We reported values herein as the mean $\pm$ standard error.

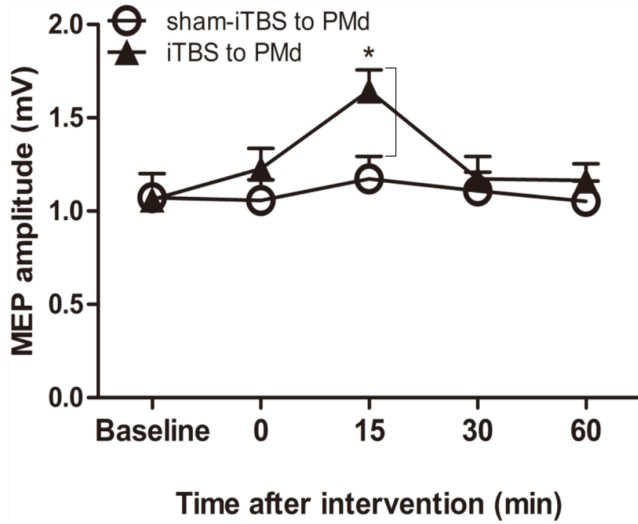

**Figure 2** **Effect of intermittent theta burst stimulation (iTBS)-induced dorsal premotor cortex (PMd) on primary motor cortex (M1) excitability.** MEP amplitudes were measured peak to peak. Circles indicate means and standard errors of the MEP amplitudes before and after sham-iTBS600 applied to the PMd interventional protocol. Black triangles indicate means and standard errors of the MEP amplitudes before and after iTBS600 applied to the PMd interventional protocol. $*p < 0.05$ compared with sham-iTBS.

## RESULTS

The measurements of MEP amplitude, SICI, ICF did not show significant differences (all $p > 0.05$) for the different interventional protocols at baseline.

### Corticospinal M1 excitability

We examined whether the level of corticospinal M1 excitability (the final motor output) was altered when iTBS600 was applied to the PMd by measuring the change in MEP amplitude at the target FDI muscle after the application. A two-way repeated-measures ANOVA revealed a significant main effect of time ($F_{(4, 60)} = 6.767, p = 0.002$) as well as a significant interaction between interventional protocol and time ($F_{(4, 60)} = 3.431, p = 0.014$); however, the main effect of interventional protocol was not significant ($F_{(1, 15)} = 1.72, p = 0.209$). Post hoc tests indicated that compared with the application of sham-iTBS600 to the PMd, application of iTBS600 significantly increased MEP amplitude 15 min after the intervention ($p = 0.012$). Meanwhile, compared with the baseline, application of iTBS600 to the PMd significantly increased MEP amplitude in M1 15 min after the intervention ($p = 0.004$). These results suggested that corticospinal M1 excitability was significantly increased by iTBS600 applied to the PMd (Fig. 2).

### Intracortical circuits

We next examined whether application of iTBS600 to the PMd altered either the inhibitory or facilitatory intracortical M1 circuits by measuring SICI and ICF, respectively, as determined by the MEP amplitude change in the target FDI muscle evoked by paired-pulse stimulation. For SICI, A two-way repeated-measures ANOVA found that both

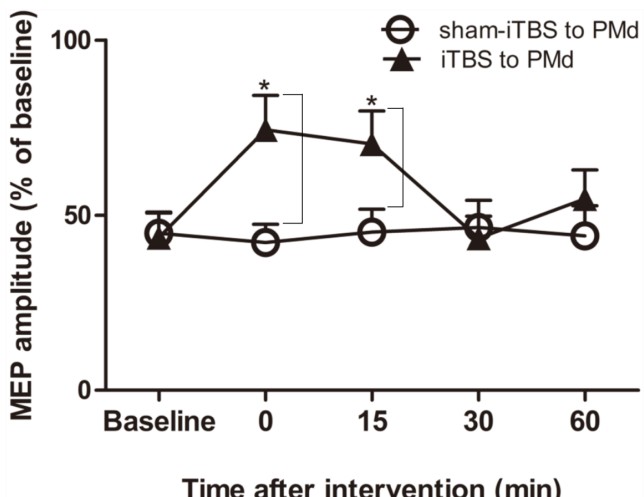

**Figure 3** **Effects of intermittent theta burst stimulation (iTBS) applied to the dorsal premotor cortex (PMd) on the inhibitory intracortical circuits of the primary motor cortex.** The effect on short-interval intracortical inhibition (SICI) was investigated by determining the conditioned motor-evoked potential (MEP) amplitudes expressed as the percentage of the MEP amplitudes induced by the test stimulus at baseline alone. Circles indicate means and standard errors of the SICI before and after sham-iTBS600 applied to the PMd interventional protocol. Black triangles indicate means and standard errors of the SICI before and after iTBS600 applied to the PMd interventional protocol. *$p < 0.05$ compared with sham-iTBS.

the main effect of time ($F_{(4,60)} = 3.247$, $p = 0.018$) and the interaction between time and interventional protocol ($F_{(4,60)} = 5.257$, $p = 0.001$) were significant, whereas the main effect of interventional protocol was not ($F_{(1,15)} = 3.016$, $p = 0.103$). Post hoc tests indicated that compare with the application of sham-iTBS600 to the PMd, application of iTBS600 to the PMd significantly decreased SICI in M1 at 0 min ($p = 0.014$) and at 15 min ($p = 0.037$) after the application, suggesting that iTBS600 applied to the PMd decreased inhibitory intracortical MI circuits (Fig. 3).

For assessment of ICF, a two-way repeated-measures ANOVA found that neither the main effect of time ($F_{(4,60)} = 1.298$, $p = 0.281$) nor of interventional protocol ($F_{(1,15)} = 0.906$, $p = 0.356$) was significant. However, the interaction between time and interventional protocol was significant ($F_{(4,60)} = 3.087$, $p = 0.022$). Post hoc tests indicated that compared with sham-iTBS600 applied to the PMd, iTBS600 applied to the PMd significantly increased ICF in M1 at 15 min ($p = 0.033$) after the application, suggesting that iTBS600 applied to the PMd increased activity at facilitatory intracortical MI circuits (Fig. 4).

## DISCUSSION

In the present study, we used rTMS to examine whether the excitability of intracortical circuits in M1 were changed by the aftereffects of iTBS600 applied to the PMd. Our primary findings were that compared with sham-iTBS600 applied to the PMd, 15 min after iTBS600 was applied to the PMd (to induce alterations in cortical excitability beyond a relatively
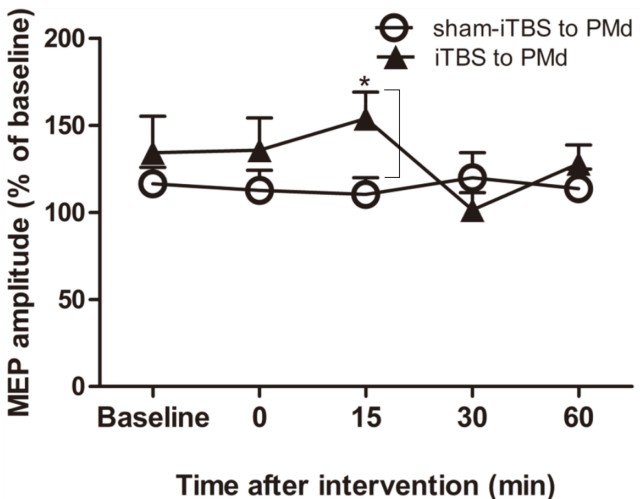

**Figure 4** **Effects of intermittent theta burst stimulation (iTBS) applied to the dorsal premotor cortex (PMd) on faciliatory intracortical circuits of the primary motor cortex.** The effect on intracortical facilitation (ICF) was investigated by determining the conditioned motor-evoked potential (MEP) amplitudes expressed as a percentage of the MEP amplitudes induced by the test stimulus at baseline alone. Circles indicate means and standard errors of the ICF before and after application of the sham-iTBS600 interventional protocol to the PMd. Black triangles indicate means and standard errors of the ICF before and after the iTBS600 interventional protocol was applied to the PMd. *$p < 0.05$ compared with sham-iTBS600.

short stimulation period), corticospinal M1 excitability measured via the change in MEP amplitude was significantly higher, and the degree of the facilitation among the intracortical circuits measured via a change in ICF was significantly increased. In addition, immediately as well as 15 min after this iTBS600 application, the degree of the inhibition among the intracortical circuits (intrinsic to the motor cortex) measured via a change in SICI was significantly decreased.

## Neurophysiology of premotor–motor connections

Synaptic activity between interconnected brain regions is altered even at rTMS intensity levels lower than the cortical threshold intensity (*Bestmann et al., 2003*). A previous study has shown that iTBS applied to M1 increases MEP amplitude at the target muscle for approximately 15 min (*Huang et al., 2005*). Diverse stimulation protocols may involve long-term potentiation (LTP)-like and long-term depression (LTD)-like processes not only in the motor cortex but also in non-motor cortex in vivo (*Chung et al., 2017*). Previous studies have highlighted the role of N-methyl-D-aspartate (NMDA) receptors as key mediators of excitability changes and excitatory synaptic transmission in the brain, which could explain the aftereffects of TBS on neuronal circuitry (*Huang et al., 2005*). Application of this model to our results would suggest that iTBS modulation of the PMd enhanced its excitability and induced an LTP-like effect as well as an increase in MEP amplitude and a decrease in SICI in M1. The increase in MEP amplitude and decrease in SICI in M1 suggested that the effectiveness of the synaptic connections was increased by iTBS applied to the PMd, which induced an LTP-like effect. These changes

are consistent with increased activation of NMDA receptors in the PMd and decreased gamma-aminobutyric acid (GABA)ergic levels in M1 and suggest that LTP-like effects in the PMd may be associated with LTD-like effects in M1. At the synaptic level, the fine balance between excitation (mediated by glutamate) and inhibition (mediated by GABA) is crucial for optimal neuroplasticity (*Dai et al., 2016*). In the present study, increased MEP amplitude was observed 15 min after the iTBS protocol was applied, whereas SICI decreased immediately after iTBS. The speeds at which the various receptors involved activate their effectors and the amounts of the neurotransmitters released may have contributed to these results. Excitatory interneurons receive inputs from inhibitory interneurons that mediate SICI (*Ni et al., 2011*), with many connections organized in a center–surround pattern to initiate point to point facilitation and widespread inhibition. Such an arrangement would suggest more possibilities for transmission from inhibitory rather than facilitatory receptors, leading to an earlier plasticity change (*Hanajima et al., 2001*).

### iTBS aftereffects on intracortical circuits

During TMS, the MEP amplitude at the muscle is determined by M1 output. This output is a complex product of interactions between excitatory facilitation and inhibitory actions of the stimulated region as well as networks within M1 itself and other motor-related cortical areas connected with M1. In the present study, the decrease in SICI and increase in ICF observed after iTBS of the PMd suggested that the effectiveness of the synaptic connections increased in M1. Previous work has shown that enhanced $GABA_A$ receptor activation increases SICI (*Ziemann et al., 1996*). Given this finding, the decreased SICI observed in the present study could reasonably be attributed to activation of $GABA_A$ receptors. The final output relies on the interplay between the inhibitory inputs and excitatory inputs projected onto the corticospinal neurons. Thus, the observed increased M1 activity may be associated with an iTBS-induced increase in postsynaptic activity, similar to that related to the LTP that is also mediated by $GABA_A$ receptors. However, TBS at intervals of 200 ms (5 Hz theta rhythm) has been shown to promote LTP by decreasing inhibitory input, enabling larger NMDA receptor responses both presynaptically and postsynaptically (*Davies et al., 1991*; *Thickbroom, 2007*). Such a trans-synaptic modulation of the iTBS-stimulated PMd would greatly contribute to the later effects we observed (at 15 min) at the PMd projection site of M1. Thus, a TBS-induced presence of GABAergic activity at the intracortical level may cause a change in the excitability of another brain region without any stimulation of that latter region.

### Enhancement of motor abilities through noninvasive stimulation

The PMd was selected for this study because it is important for the precise timing and execution of unimanual and coordinated bimanual movements (*Ni et al., 2009*). Although the present study focused solely on the physiological data, our data offer an undergirding mechanism for future work involving behavioral-level analyses to advance understanding of motor learning and control.

Previous research has indicated that 50-Hz triple-pulse rTMS of the PMd increases both motor function and motor learning; for example, the ability to learn and to perform

a motor sequence task or a skilled grasping task is increased (*Gregori et al., 2005*). TMS applied over the left PMd has been shown in one study to decrease the lack of grip strength as well as to inhibit corticospinal excitability. The authors of that study suggested that the effect of blocking the PMd preparatory activity indicated a causal role for the PMd in grip force (*Duque et al., 2010*). Future investigations will be needed to clarify how TBS is associated with motor learning at the behavior level.

Our study was not without limitations that may affect the interpretation of our results. A main limitation was the absence of a neuronavigation system to accurately localize the target areas during the experimental sessions. A second limitation was that we used a relatively small number of participants who were close in age, restricting the generalizability of the results to younger or older individuals. Therefore, future research should aim for a larger number of participants and different age groups.

## CONCLUSIONS

The present study provided evidence in support of our hypothesis that the application of iTBS to the PMd increases excitability and decreases SICI in M1. These results suggested that a facilitatory network exists from the PMd to the M1. The existence of such a network informs the understanding of the allocation of resources to achieve optimal motion output.

### Funding
The present study was funded by the Outstanding Clinical Discipline Project of Shanghai Pudong (PWYgy2018-04). The funders had no role in study design, data collection and analysis, decision to publish, or preparation of the manuscript.

### Grant Disclosures
The following grant information was disclosed by the authors:
Outstanding Clinical Discipline Project of Shanghai Pudong: PWYgy2018-04.

### Competing Interests
The authors declare there are no competing interests.

### Author Contributions
- Hai-Jiang Meng and Na Cao conceived and designed the experiments, performed the experiments, analyzed the data, prepared figures and/or tables, authored or reviewed drafts of the paper, and approved the final draft.
- Jian Zhang and Yan-Ling Pi conceived and designed the experiments, authored or reviewed drafts of the paper, and approved the final draft.

### Human Ethics
The following information was supplied relating to ethical approvals (i.e., approving body and any reference numbers):

The protocol was approved by the ethics committee of the Sport University of Shanghai (SUS2014024).

## Data Availability

The raw measurements are available in the Supplemental File.

## Supplemental Information

Supplemental information for this article can be found online at http://dx.doi.org/10.7717/peerj.9253#supplemental-information.

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
