# Peer review of "Intermittent theta burst stimulation facilitates functional connectivity from the dorsal premotor cortex to primary motor cortex"

_PeerJ, doi:10.7717/peerj.9253_

## Round 0.1 · original submission · Major Revisions

Two reviewers have now read your manuscript and have requested major revisions. Please address each of these line-by-line in your revision response letter before resubmitting.

Reviewer 1 ·

Basic reporting

The topic of the manuscript is of sufficient interest, the article is well structured but the figures are of poor quality. Results are correctly reported and sufficiently well discussed from a physiological point of view. The main problem of the manuscript is the language, which presents several mistakes and is not fluently. I strongly recommend an english revision from a native speaker.

Experimental design

The aim of the study is well defined and quite relevant, although the novelty of the study is low. The methods are well described with sufficient details to reproduce the experiment.

A main limitation of the study is the absence of a neuronavigator system to localize the target areas. This is a weak point especially as regards the localization of PMd

Validity of the findings

The experimental designed is well constructed with a sham condition of sufficient validity. Statistics are well planned. The conclusions are well stated and sufficiently support the results of the study.

Additional comments

The manuscript is well designed and the topic is of sufficient interest, although the novelty is not high. The main limitation of the work is the language writing. I strongly recommend a revision by a native english speaker. In addition there are some sentences throughout the manuscript that need revision, for instance:

line 91-12 "TMS applied...M1 regions". This sentence does not have a scientific validity. What kind of TMS should strength M1 connectivity?

line 246-247 "decrease in SICI...connections decreased". The physiological meaning of this sentence is not clear to me.

line 267-268 "PMd rTMS...learning". What kind of rTMS are the authors talking about? Please be more precise.

Reviewer 2 ·

Basic reporting

In my opinion this manuscript suffers of some critical points. I have several suggestions that the authors may want to take into consideration.
In first instance, the main concern regards the small number of participants, that allows to judge the results, included the performed correlation, only as very preliminary. At this regard, I suggest to collect additional data for a better validation of the findings. Regarding the introduction, it suffers from too general statements regarding the TMS protocols and the cortical changes induced in terms of plasticity changes, lacking of any specific explanation of the neurophysiological mechanisms for which TMS approach could be useful to evaluate the PMd-M1circuitry.
Additionally, it lacks of any relevant reference about the relationship between premotor and motor cortices (for example, Koch G, et al. J Physiol 2007. Interactions between pairs of transcranial magnetic stimuli over the human left dorsal premotor cortex differ from those seen in primary motor cortex).

Experimental design

From a methodological point of view, the manuscript lacks of a clear hypothesis. What was the rationale for the choice of a specific interstimulus interval (i.e., 2 ms) for SICI protocol? Why the authors did not consider to assess both inhibitory and facilitatory mechanisms, by means of a short intracortical inhibition/facilitation (SICI-ICF) protocol? Did the authors used a neuronavigation system, for a correct targeting of TMS pulses during the different experimental sessions (before and after both experimental interventions, i.e. iTBS600 and sham)?

Validity of the findings

The discussion did not report a clear explanation about the possible neurophysiological mechanisms underlying the cortico-cortical interaction between PMd and M1. Could the authors explain in more detail their results, also regarding the time-points in which the main findings were observed?

Additional comments

In this study, Meng and colleagues aimed to investigate the influence of dorsal premotor cortex on motor cortical activity, by applying a theta burst stimulation protocol (intermittent TBS600 vs sham) over PMd and by assessing motor cortical excitability and intracortical circuits over M1. The results highlighted an increase of motor evoked potentials and a decrease in SICI at specific time-points after intermittent TBS as compared with sham.
In my opinion this manuscript suffers of some critical points. I have several suggestions that the authors may want to take into consideration.
In first instance, the main concern regards the small number of participants, that allows to judge the results, included the performed correlation, only as very preliminary. At this regard, I suggest to collect additional data for a better validation of the findings.
Regarding the introduction, it suffers from too general statements regarding the TMS protocols and the cortical changes induced in terms of plasticity changes, lacking of any specific explanation of the neurophysiological mechanisms for which TMS approach could be useful to evaluate the PMd-M1circuitry.
Additionally, it lacks of any relevant reference about the relationship between premotor and motor cortices (for example, Koch G, et al. J Physiol 2007. Interactions between pairs of transcranial magnetic stimuli over the human left dorsal premotor cortex differ from those seen in primary motor cortex).
From a methodological point of view, the manuscript lacks of a clear hypothesis. What was the rationale for the choice of a specific interstimulus interval (i.e., 2 ms) for SICI protocol? Why the authors did not consider to assess both inhibitory and facilitatory mechanisms, by means of a short intracortical inhibition/facilitation (SICI-ICF) protocol? Did the authors used a neuronavigation system, for a correct targeting of TMS pulses during the different experimental sessions (before and after both experimental interventions, i.e. iTBS600 and sham)?

The discussion did not report a clear explanation about the possible neurophysiological mechanisms underlying the cortico-cortical interaction between PMd and M1. Could the authors explain in more detail their results, also regarding the time-points in which the main findings were observed?

---

## Round 0.2 · Major Revisions

The reviewers have read your revised manuscript and are basically happier now. Rev 1 has some very good suggestions with respect to the statistical analysis: this is the purpose of the “major revisions” decision. I imagine a few additional analyses and clarifications will clear this up. Everything else is minor and should be easy for you to fix.

Reviewer 1 ·

Basic reporting

The introduction is well written and provide a sufficiently clear explanation of the rationale. However, it would be appropriate to mention new TMS-EEG study that demonstrated that TBS affects not only the site of stimulaton but also interconnected motor and non-motor areas.
Other minor points:

line 58-59: this sentence is too general since the authors should be more clear on which kind of "stimulation" are they referring to, especially when they refer to structural changes of the cortex.

line 79-80: this is inaccurate, since TMS stimulate not only facilitatory interneurons but also inhibitory

Experimental design

The experimental design presents a main point of weakness, namely the authors did not use any neuronavigation system to monitor coil position nor to identify the PMd site of stimulation.
In addition I have several concernes about the experimental procedures:
1. How did the authors establish the number of experimental participants? Did they perform a power analysis?

2. As far as I have understood, the authors re-tested 1mV intensity at each time point. To me, this procedure is not accurate and could lead to some bias. Can the authors better explain this procedure providing some references to justify this choice?

3. I find not convincing the choice to use only one ISI for SICI and ICF. The authors justify this choice referring to some preliminary test however they do not report this data.

4. The statistical section is poorly described. Which factors were included in the ANOVAs? Were the data tested for normality and sphericity?

Validity of the findings

Corticospinal M1 excitability results:
1. As a preliminary analysis, did the authors tested for difference in baseline between the two conditions? This should be explicited.

2. The authors reported a significant time*protocol interaction. However, I do not completely understand which post-hoc tests were detected as significant. The authors reported only one significant difference on MEPs tested at 15 minutes between the two protocols. However, it is not clear whether MEPs testeat at 15 min in the TBS condition were different also compared to the other time points of the same condition. This is important to understand if iTBS effectively induce an LTP-like effect. If not, this should be reported in the results and discussed.

3. Generally speaking, by looking at the results figures, I do not find easy to understand which differences are significant.

4. The correlation analysis is not convincing. First, it seems that the significant correlation is led by a few cases. Second, it is not clear how many correlations did the authors run and if a correction for multiple correlations was applied.

Additional comments

The study is well-written and the physiological interpretation of the findings is convincing. However, the experimental design and the results suffer from several methodological limitation that I explained in detail in the previous section.

Reviewer 2 ·

Basic reporting

No comment

Experimental design

No comment

Validity of the findings

No comment

Additional comments

The authors responded satisfactorily to all my comments.
Please correct "faciliatory" with "facilitatory" (lines 53, 59, 124,215, 238, 306, 356)

---

## Round 0.3 · accepted · Accept

Thank you for your revisions! We are happy to now accept your article.